# The Spatial Mechanism and Predication of Rural Tourism Development in China: A Random Forest Regression Analysis

Xishihui Du [1], Zhaoguo Wang [2,*] and Yan Wang [1]

1   School of Transportation and Geomatics Engineering, Shenyang Jianzhu University, Shenyang 110168, China; daisy_duxi@126.com (X.D.); wyan413@163.com (Y.W.)
2   College of Economic and Management, Shenyang Agricultural University, Shenyang 110866, China
*   Correspondence: wzglinyi2007@163.com

**Abstract:** Rural tourism has long been recognized as a significant strategy for promoting rural revitalization in China. Excessive development has had a number of negative consequences for rural tourism. As a result, there is a growing need to optimize the developmental framework of rural tourism in order to ensure its sustainable growth. This study focuses on key tourism villages and employs geostatistical analysis and the random forest methodology to elucidate the spatial mechanisms underlying rural tourism and identify potential areas for its development in China. The research findings reveal several important insights: (1) Key tourism villages exhibit a concentrated spatial distribution, characterized by pronounced regional disparities. (2) The intrinsic characteristics of rural areas and the conditions conducive to tourism development play pivotal roles in shaping rural tourism. Notably, cultural resources, tourism resources, rural accessibility, and tourism potential are identified as the primary influential factors. (3) Predictive modeling using random forest analysis indicates that densely populated areas in the eastern region retain the highest level of suitability for rural tourism. In contrast, the development of rural tourism in western and border regions encounters certain constraints. Additionally, the northern region encompasses larger expanses with high suitability, whereas the southern region is generally moderate. This comprehensive nationwide investigation provides valuable insights into the key aspects of rural tourism development and offers practical guidance for achieving sustainable rural tourism practices in China.

**Keywords:** key rural tourism villages; spatial mechanism; random forest; China

## 1. Introduction

In recent times, tourism has been considered a viable path to sustainable development, given its potential to bring in significant foreign exchange earnings, generate local employment opportunities, and promote economic growth [1–3]. The impact of urbanization has progressively led to rural tourism becoming a popular trend [4,5]. Rural tourism sustains its prominent position in the tourism industry, as visitors increasingly prioritize the quest for health and wellness in the post-COVID-19 era [6,7].

Rural tourism destinations are of significant interest to the academic community [3,8–11]. The research focus on rural tourism concentrates on its contribution to the development of rural areas and the transformative effects it has on these regions, often through the lens of case studies [7,12,13]. The rapid growth of rural tourism has led to an increasingly competitive market environment, which has in turn created a need for scientific advancement in the field [14]. Scholars have increasingly turned their attention towards understanding the spatial distribution patterns of its development and exploring the underlying spatial mechanisms driving these patterns [15]. The intricate and multifaceted spatial distribution patterns observed in rural tourism villages arise from the diverse interplay of numerous influencing factors [16]. Hence, it is essential to conduct a thorough investigation that encompasses the overall spatial arrangement of rural tourism villages, while also considering

the specific contextual factors that influence their spatial distribution. This comprehensive study is crucial for unraveling the complex interplay of variables and understanding the nuanced dynamics that shape the spatial patterns of rural tourism.

The Seventh National Census of Population, conducted in 2020, revealed that a substantial proportion of China's population, exceeding one-third, continues to reside in rural areas. Thus, the attention towards economic development and living conditions in rural areas has been a longstanding issue of concern [17,18]. The shifting market trends have triggered significant changes in the industrial composition of numerous rural regions in China. Consequently, new sectors have emerged, with rural tourism taking center stage as a prevailing mechanism for rural regeneration and conservation. Operating as a multi-industrial sector, rural tourism drives comprehensive economic, social, and spatial transformations and reconstructions within rural areas [3,19–21]. With its ability to stimulate industrial integration, enhance the value of agricultural products, augment farmers' earnings, and reinforce the foundation of rural collective economy, rural tourism can facilitate notable advancements [22]. In 2018, China introduced the Rural Revitalization Strategy, and the role of rural tourism in stimulating the rural revitalization has since obtained renewed attention [18].

In 2019, the Ministry of Culture and Tourism and the National Development and Reform Commission jointly initiated a national campaign to identify key villages for promoting rural tourism. The screening of key rural tourism villages (key villages) is a multi-level process that begins at the municipal level. Municipal cultural and tourism bureaus first identify villages that have a high concentration of tourism resources, a diversity of tourism products, comprehensive tourism facilities, and attractive rural settings. These villages are then recommended to the provincial cultural and tourism department, and then the National Ministry of Culture and Tourism conducts a rigorous evaluation to determine the final list of key tourism villages. Once selected, these villages receive significant support from the government, including financial aid and training opportunities. As of 2022, the lists of four batches have been released for public reference, and designations for key villages reached a total of 1399. These designated villages serve as exemplary paradigms for advancing the development of high-quality rural tourism [23], contributing to the optimization of rural tourism supply, and spearheading the growth of the rural tourism industry. As a result, these key villages have assumed a pivotal role in examining the progress and transformative dynamics within the domain of rural tourism in China.

This research investigates the spatial mechanisms underlying key villages in China and identifies the principal factors influencing their development. It employs spatial statistic methodologies to analyze data derived from the key rural tourism villages catalog. This study provides a comprehensive overview of rural tourism in China, and offers innovative insights into the sustainable advancement of rural tourism.

The subsequent sections of this article are structured as follows: Section 2 provides a brief review of spatial statistical research on rural tourism in China. Section 3 presents a comprehensive overview of the research design and analytical approach utilized in this study. Section 4 examines the spatial configuration of key villages, offering detailed insights into the factors and associated mechanisms that exert influence on these rural tourism destinations and enable the prediction of suitable tourism zones. Section 5 provides a concise discussion of the preceding sections. Lastly, the implications of the findings for future research and limitations of the research are summarized in the conclusion.

## 2. Literature Review

Scholars in China have conducted extensive analyses on the spatiotemporal distribution patterns of rural tourism, with particular emphasis on notable instances such as beautiful leisure villages [24], traditional villages [25], and key villages of rural tourism [16,26]. These analyses have been conducted at multiple levels, including national [16,24,25], regional [26,27], and provincial [10,28], with the objective of elucidating the underlying determinants that influence these distributions. The distribution of rural tourism villages exhibits a pronounced spatial imbalance, characterized by a notable concentration of such villages in the eastern region of the country, predominantly positioned southeast of the Hu line [29]. Furthermore, the density of these villages also demonstrates an apparent inter-provincial variation [10]. The resource endowment premise and environmental conditions play a decisive role in defining the geographical distribution of rural tourism destinations [16,25]. Moreover, the spatial characteristics and structural arrangements of rural tourism villages are subject to the influence of multiple factors. These factors include socio-economic elements such as economic status, transit accessibility, tourist demand, policy orientation, and service quality. Additionally, the availability of natural and cultural resources, as well as geographical factors, play significant roles. The influence exerted by these factors varies in degree, shaping the spatial patterns and organizational models observed in rural tourism villages [29,30]. Despite these valuable findings, empirical research on the spatial mechanism shaping rural tourism remains limited, warranting further attention and investigation.

By contrast, machine learning represents a subset of artificial intelligence where computers acquire the ability to learn autonomously, without explicit programming. By utilizing statistical techniques, machine learning algorithms facilitate data analysis to identify patterns, enabling the generation of predictions or decisions [31]. Thereinto, the random forest (RF) operates as an ensemble learning method, constructing a multitude of decision trees during the training phase, and subsequently determining the class that represents the mode of the classes (for classification) or the mean prediction (for regression) of the constituent trees [32–34]. The RF algorithm, as an ensemble learning technique, relies on the combination of a large number of decision trees. Each tree is trained using a random selection of variables and a random subsample from the training dataset [35]. The RF methodology has demonstrated successful utilization in both quantifying the factors influencing various phenomena and estimating the spatial distributions of homestays in Beijing [36]. Hence, the RF possesses the advantage of being able to evaluate the relative significance of independent variables on dependent variables [37].

## 3. Methodology

### 3.1. Research Design

This study employs geospatial statistics methodology to explore the spatial characteristics and investigate the factors influencing key villages in China. In addition, the RF algorithm is integrated in this investigation as a powerful analytical tool. The research adheres to a systematic framework encompassing three primary steps. In Step 1, the identification and analysis of key villages are undertaken, utilizing ArcGIS Pro 3.0 to discern their spatial patterns. Step 2 involves the compilation and organization of spatial geographic data, incorporating raster data representing influential factors, and examining their spatial associations with village density. Lastly, Step 3 incorporates the application of the RF algorithm to unveil the underlying spatial mechanism governing the distribution of key villages and predict the potential for rural tourism development in China. The system architecture employed to elucidate this spatial mechanism is illustrated in Figure 1.

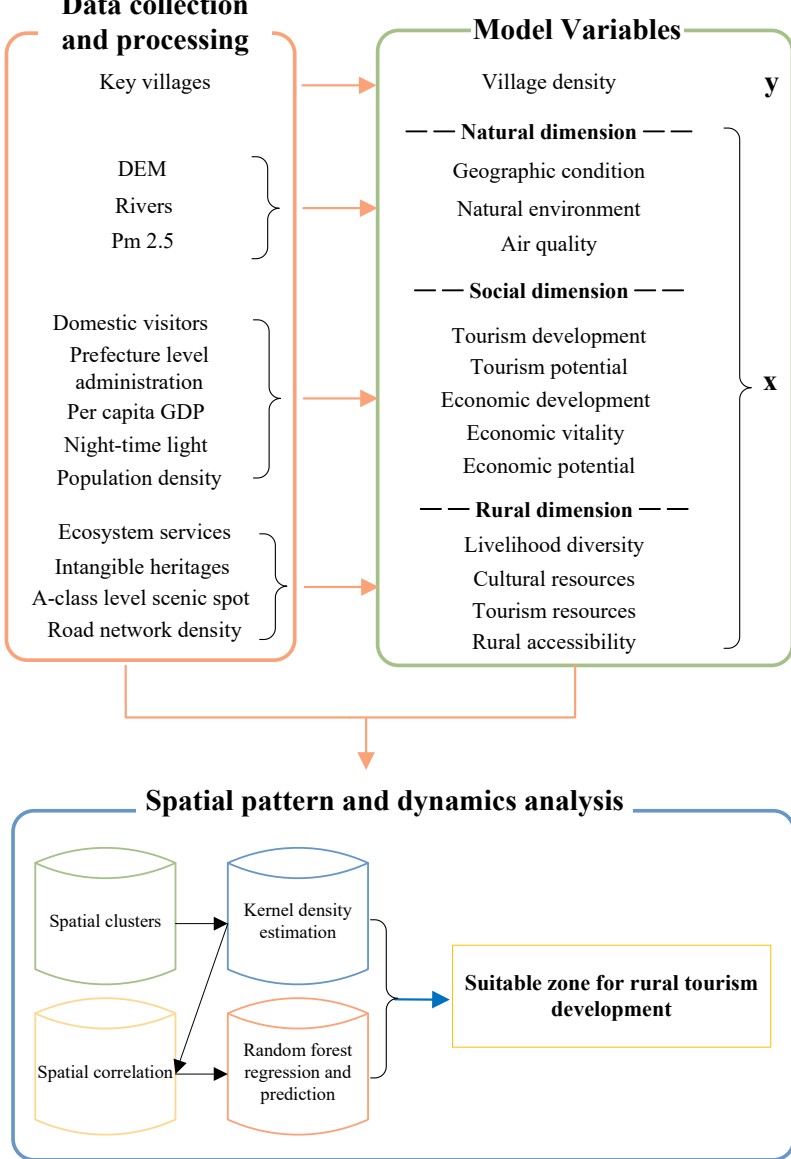

**Figure 1.** Overview of the research framework.

*3.2. Data*

3.2.1. Key Rural Tourism Villages

Research was conducted on key villages published on the website of the Ministry of Culture and Tourism of the People's Republic of China (https://www.mct.gov.cn, accessed on 1 March 2023). The coordinates of the 1399 key villages were obtained using the Baidu Map application programming interface (API) coordinate selection system [10]. The specific coordinates were then used to visualize the locations of the key villages on the map via ArcGIS Pro 3.0 (Figure 2), and the key rural tourism villages showed a trend of aggregated distribution in space.

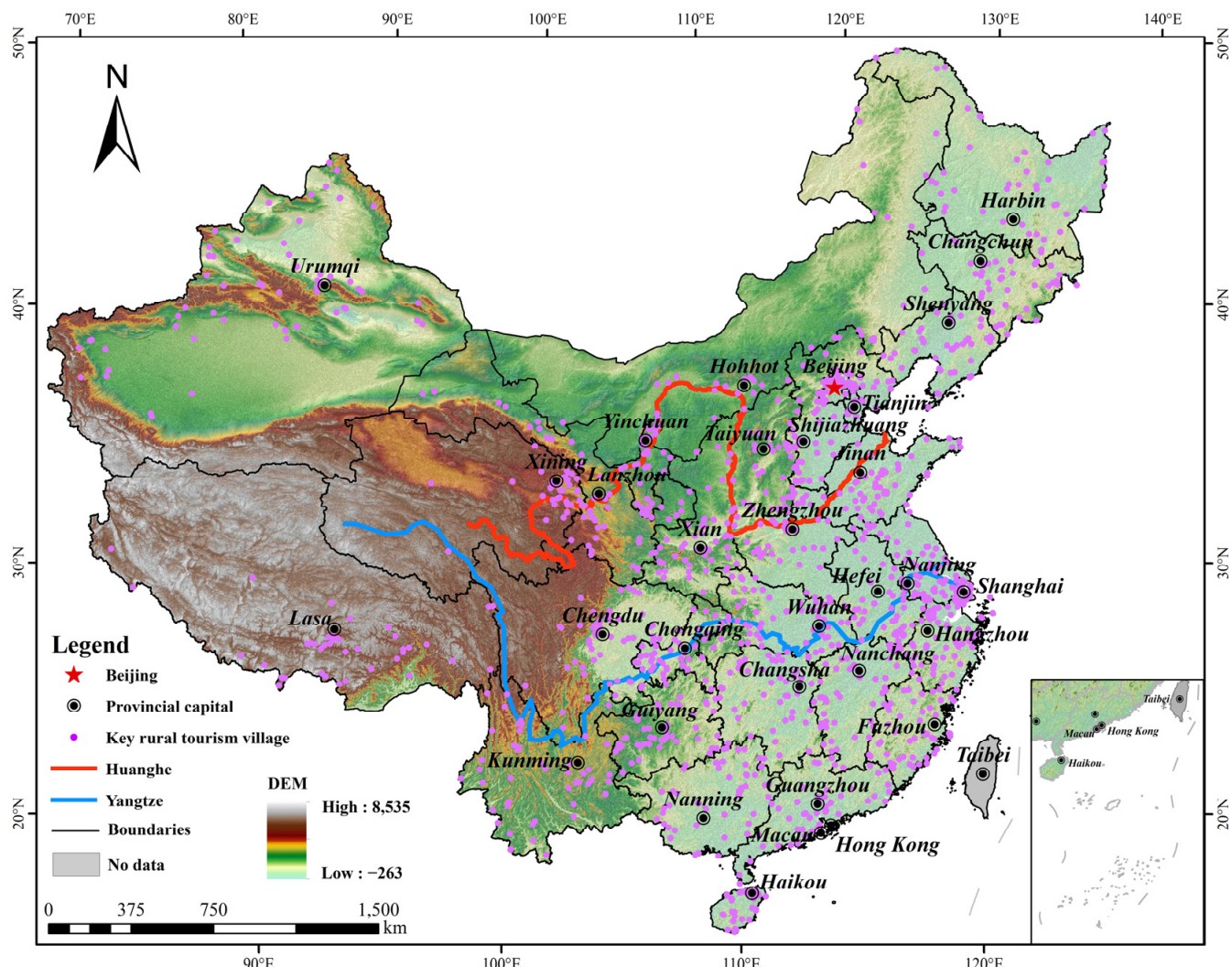

**Figure 2.** Overview of key rural tourism villages in China.

### 3.2.2. Influential Factors

Rural tourism represents a multifaceted social phenomenon that is shaped by numerous factors, encompassing elements of nature, society, and the rural context. Key tourism villages are a manifestation of policy orientations, reflecting the government's priorities and goals for rural tourism development. Considering the existing body of research and data availability, 12 influential driving factors were carefully chosen and identified for this study. These factors, presented in Table 1, have been selected based on their significance and relevance to the study's objectives, providing a comprehensive understanding of the complex dynamics underlying rural tourism.

Table 1 presents the variables employed in this study, categorized into natural, social, and rural elements. Under the natural elements, geographic condition (DEM), natural environment (rivers), and air quality (PM2.5) are considered. The digital elevation model (DEM) represents the topographic relief of villages, influencing their spatial distribution [25]. Rivers serve as vital natural features, providing water, food, and transportation, which are fundamental for rural tourism development and essential for human sustenance and productivity. PM2.5, a form of air pollution comprising microscopic particles, poses risks to human health. Rural areas characterized by favorable environmental and air quality conditions emerge as the optimal choice for urban residents seeking recreational vacations [5].

**Table 1.** The influential factors of key rural tourism villages.

| Dimension | Factors | Data | Source | Reference |
|---|---|---|---|---|
| Natural elements | Geographic condition | DEM | Resource and Environment Science and Data Center (https://www.resdc.cn/, accessed on 1 May 2023) | [27,29] |
| | Natural environment | Rivers | National Geomatics Center of China (http://www.ngcc.cn/ngcc/html/1/391/392/16114.html, accessed on 1 May 2023) | [26] |
| | Air quality | PM2.5 | Atmospheric Composition Analysis Group (https://sites.wustl.edu/acag/datasets/surface-pm2-5/, accessed on 1 May 2023) | [38,39] |
| Social elements | Tourism development | Domestic visitors | China's economic and social big data research platform (https://data.cnki.net/, accessed on 1 May 2023) | [29,40] |
| | Tourism potential | Prefecture-level administration | National Geomatics Center of China (http://www.ngcc.cn/ngcc/html/1/391/392/16114.html, accessed on 1 May 2023) | [41] |
| | Economic development | Per capita GDP | Resource and Environment Science and Data Center (https://www.resdc.cn/, accessed on 1 May 2023) | [16,25] |
| | Economic vitality | Night-time light | | [38] |
| | Economic potential | Population density | | [41,42] |
| Rural elements | Livelihood diversity | Ecosystem services | | [38,40] |
| | Culture resources | Intangible heritage | Chinese Intangible Cultural Heritage website (https://www.ihchina.cn/, accessed on 23 April 2023) | [10,16,25,29] |
| | Tourism resources | A-class scenic spot | Ministry of Culture and Tourism of China (https://www.mct.gov.cn/, accessed on 20 April 2023) | [11,29,40,43] |
| | Rural accessibility | Road network | National Geomatics Center of China (http://www.ngcc.cn/ngcc/html/1/391/392/16114.html, accessed on 1 May 2023) | [44–46] |

The social elements comprise tourism development (domestic visitors), tourism potential (prefecture-level administration), economic development (per capita GDP), economic vitality (night-time light data), and economic potential (population density). Regional economic development plays a pivotal role in shaping the distribution, growth, and establishment of key rural tourism villages. Domestic visitors reflect the preference for short, frequent, and convenient vacation options [40]. Prefecture-level administration denotes the tourism potential influenced by this market scale. Per capita GDP indicates the economic output per individual within a specific area. Night-time light data provides insights into economic vitality, representing the amount of nocturnal illumination emitted from a given region. Population density, the number of individuals per unit area, influences the spatial arrangement of key rural tourism villages, with higher population densities correlating with larger tourism market capacities and potential.

The rural elements encompass livelihood diversity (ecosystem services), cultural resources (intangible heritage sites), tourism resources (A-class scenic spots), and rural accessibility (road network density). Ecosystem services encompass the benefits derived from ecosystems, such as clean air, water, food provision, and recreational opportunities. Intangible heritage sites represent cultural assets that are non-physical, encompassing traditional knowledge and practices. China boasts the largest number of intangible cultural heritage items globally [16]. Tourism resource endowment plays a crucial role in

influencing the tourism development of a region [24]. A-class scenic spots refer to top-tier tourist destinations designated by the government. The source market for rural tourism primarily consists of neighboring cities, emphasizing self-driving tours. Rural accessibility, specifically road network density, facilitates transportation and acts as one of the essential elements connecting tourism destinations with tourist sources.

Multisource data were obtained to reflect the index in a qualitative way. As such, population density and per capita GDP were obtained from the Resource and Environment Science and Data Center to a precision of 1 km. The initial data of these factors exhibited heterogeneity with respect to data format and resolution, necessitating the use of ArcGIS 10.8 software to create a standardized 30 × 30 km fishnet grid based on the mean observation distance of key villages. The objective of this procedure was to establish consistency in terms of the coordinate system (Krasovsky_1940_Albers) and resolution of the data. Each vector factor was assigned a categorical value ranging from 1 to 5, with higher values representing superior grades and lower values indicating lower grades. Subsequently, kernel density interpolation was applied to generate national-scale maps for each factor. The resulting maps were then linked to the fishnet data using ArcGIS Pro 3.0, enabling the acquisition of geospatial data necessary for investigating the spatial mechanism of rural tourism. To visually represent the varying degrees of each factor, the Jenks natural breaks method was employed (Figure 3).

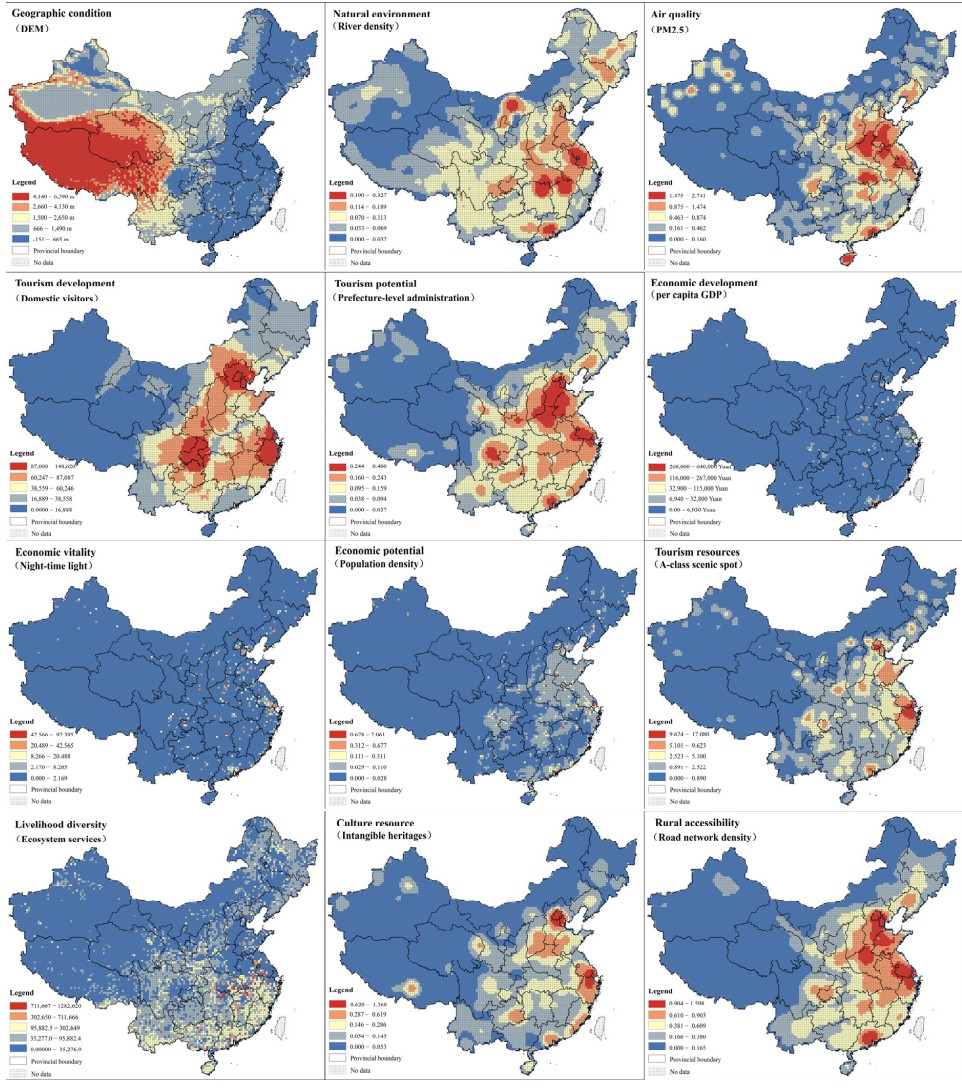

**Figure 3.** Influential factors of the distribution of key rural tourism villages.

### 3.3. Methods

3.3.1. HDBSCAN (Hierarchical Density-Based Spatial Clustering of Applications with Noise)

The HDBSCAN algorithm, developed by Campello, Moulavi, and Sander in 2013 [47], offers an extension to the DBSCAN algorithm by introducing a hierarchical clustering framework and incorporating a mechanism to extract a flat clustering based on cluster stability. HDBSCAN is a density-based clustering algorithm that is well suited to extracting clustering results at different density levels [48,49]. This is because HDBSCAN identifies clusters based on the density of points, and it can identify both dense clusters and noise.

In ArcGIS Pro 3.0, the Density-Based Clustering tool utilizes the principles of HDB-SCAN to identify areas where points exhibit higher concentrations and distinguish them from areas with sparse or no points. Non-clustered points are classified as noise. It utilizes a range of distances to effectively separate clusters of different densities from sparser noise. The HDBSCAN algorithm within this tool is data-driven, requiring minimal user input and offering a more autonomous clustering approach, overcoming the limitations of DBSCAN and providing improved robustness in parameter selection.

3.3.2. Kernel Density Analysis

Density analysis is a spatial analysis technique that involves distributing known quantities of a phenomenon across a landscape, considering both the measured quantity at each location and the spatial relationship between these locations. Among various density analysis methods, kernel density analysis is widely utilized due to its ability to estimate a magnitude-per-unit area based on point or polyline features, fitting a smoothly tapered surface around each point or polyline.

The outcomes of Kernel density analysis enable the identification of concentration and dispersion patterns within the study area. Thus, the kernel density was used to analyze the spatial distribution of key villages, and thus the density of each factor was calculated. The kernel density can be calculated using the following formulation:

$$\hat{f}(x) = \frac{1}{nh} \sum_{i=1}^{n} K\left(\frac{x - x_i}{h}\right) \tag{1}$$

where $\hat{f}(x)$ represents the estimated kernel density of the villages at location $x$; $n$ is the number of villages; $h$ is the parameter that determines the smoothing level of the kernel density estimation; $K$ is the kernel function. This study used the kernel density tool integrated within the ArcGIS Pro 3.0 for mapping.

3.3.3. Bivariate Analysis

Bivariate analysis is a commonly employed statistical approach that plays a crucial role in understanding the association between two variables [22]. By utilizing scatterplots, correlation coefficients, and simple linear regression, researchers are able to visually represent and quantitatively measure the relationship between these variables. To explore the influence degree of various factors, this study used the correlation analysis method to reflect the interrelationship between the spatial distribution and the influencing factors of villages, and its formula is:

$$r = \frac{\sum_{i=1}^{n}(x_i - \overline{x})(y_i - \overline{y})}{\sqrt{\sum_{i=1}^{n}(x_i - \overline{x})^2}\sqrt{\sum_{i=1}^{n}(y_i - \overline{y})^2}} \tag{2}$$

where $r$ denotes the correlation coefficient; $x_i$ represents the density of villages for point $i$; and $y_i$ is the value of influencing factors for point i.

GeoDa offers a method for conducting exploratory data analysis, which facilitates the detection of multivariate spatial relationships. This approach enables the simultaneous examination of multiple bivariate correlations, allowing for a comprehensive analysis across

natural, social, and rural dimensions. By utilizing GeoDa, it became possible to uncover and investigate the interrelationships among various factors within these dimensions.

### 3.3.4. Random Forest Algorithm

The random forest (RF) algorithm is a machine learning technique commonly employed for fitting regression models. For prediction, each pixel's features are evaluated by each tree in the ensemble and assigned a class label [33]. The predicted pixel class is then determined by majority voting over the trees in the ensemble, and a map of potential areas for rural tourism is produced after predicting for all pixels.

In the context of this study, a dataset comprising grids containing key tourist villages was selected, resulting in 1399 sample grids. The village density was designated as the dependent variable, and the influential factors corresponding to the sample grids were considered as the feature dimensions for the RF algorithm. Through this process, the distribution of suitable rural tourism development zones could be identified.

To establish the relationships between village distribution and the natural, social, and rural elements at the spatial scale, the RF algorithm was implemented using the random forest tools integrated within ArcGIS Pro 3.0. The explanatory variables, representing the natural, social, and rural elements, were extracted from the training features. In addition to assessing the model's performance based on the training data, predictions could be made for other features as well.

## 4. Results

### 4.1. Spatial Distribution Characteristics

#### 4.1.1. Spatial Clusters of Key Rural Tourism Villages

Cluster analysis is a data-driven approach for grouping data points into clusters such that the data points within each cluster are more similar to each other than they are to data points in other clusters [49]. The current inquiry investigates the overall distribution of key rural tourism villages in China and scrutinizes their spatial distribution pattern using density-based spatial clustering analysis. Utilizing the HDBSCAN algorithm, the analysis reveals the identification of 26 distinct clusters, of which clusters 4, 2, and 1 are the top-ranking clusters, accounting for a substantial proportion (32.78%) of all key villages (Figure 4). Cluster 4, which is situated in the Yangtze Delta Plain, has traditionally functioned as the epicenter of China's economic and cultural activities. Cluster 2, on the other hand, is found in the middle and lower reaches of the Yellow River, an area well known for its historical and cultural significance. Cluster 1 is located around Beijing, the capital city of the country, which serves as its political, economic, and cultural center. Additionally, it is noted that more villages exhibit a dispersed layout and that 396 villages are deemed to be noise.

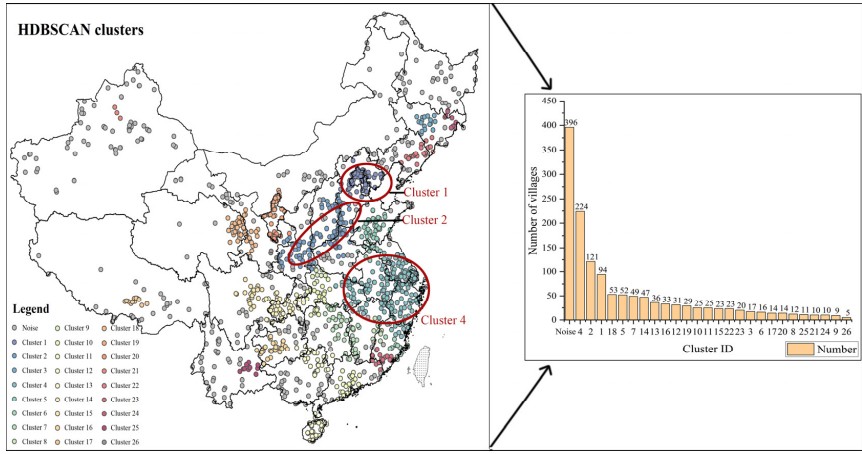

**Figure 4.** Spatial distribution clusters of key rural tourism villages in China.

### 4.1.2. Spatial Distribution Density of Key Rural Tourism Villages

The spatial distribution density of key villages in China can be further delineated in the spatial dimension through the application of kernel density analysis. The calculated density values were divided into five categories using the natural breakpoint method, and then the spatial kernel density distribution map of villages was generated (Figure 5). The spatial distribution density varied significantly in different regions. The area with low village density is concentrated at the northwest of the Hu line, with lower population. On the whole, high village density regions are concentrated around Beijing and Shanghai, with several scattered regions. This is likely due to a number of factors, such as the availability of natural resources, the presence of historical and cultural sites, and the level of economic development.

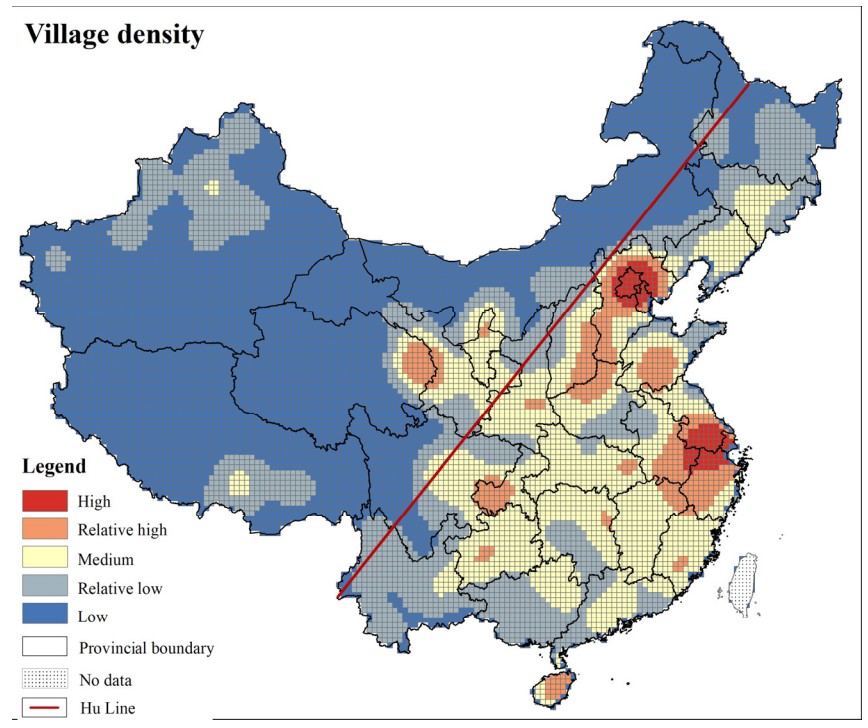

**Figure 5.** Spatial kernel density distribution of key rural tourism villages in China.

### 4.2. Spatial Interaction between Key Rural Tourism Villages and Influential Factors

In this study, village density is employed as a surrogate measure for evaluating the spatial distribution of rural villages. By employing multi-factor correlation analysis, the impact of various factors on the spatial distribution of rural villages across different dimensions are identified. This analytical approach facilitates the examination of the significance of these factors. For instance, if a positive correlation is observed between village density and natural resources, it suggests a higher likelihood of villages being situated in areas rich in natural resources.

A scatter plot matrix can effectively illustrate bivariate relationships for several variable pairings. The graph shows both positive and negative associations, as well as non-significant ones. Significance of the relationship between variables is evidenced by the linear fit slope above each scatter plot, with values of one * ($p < 0.05$) or two ** ($p < 0.01$) indicating correlation significance. The linear fit slope above each scatter plot provides correlation significance indicated as one * ($p < 0.05$) or two ** ($p < 0.01$). The histograms situated in the diagonal section of the matrix enable observation of the distribution and shape of each variable individually.

The relationship between the natural dimension and key village distribution is depicted in Figure 6. The results revealed a significant negative correlation between geographic conditions and village density ($-0.294$). Put simply, as geographic conditions

become more challenging, village density tends to decrease. Interestingly, the findings indicate a positive association between village density and air quality (0.345), specifically measured by PM2.5 levels. This implies that higher levels of PM2.5 pollutants are accompanied by increased rural tourism demand. Furthermore, a positive correlation between the natural environment, represented by the presence of rivers, and village density is observed (0.231). This suggests a tendency for villages to be situated in close proximity to river systems.

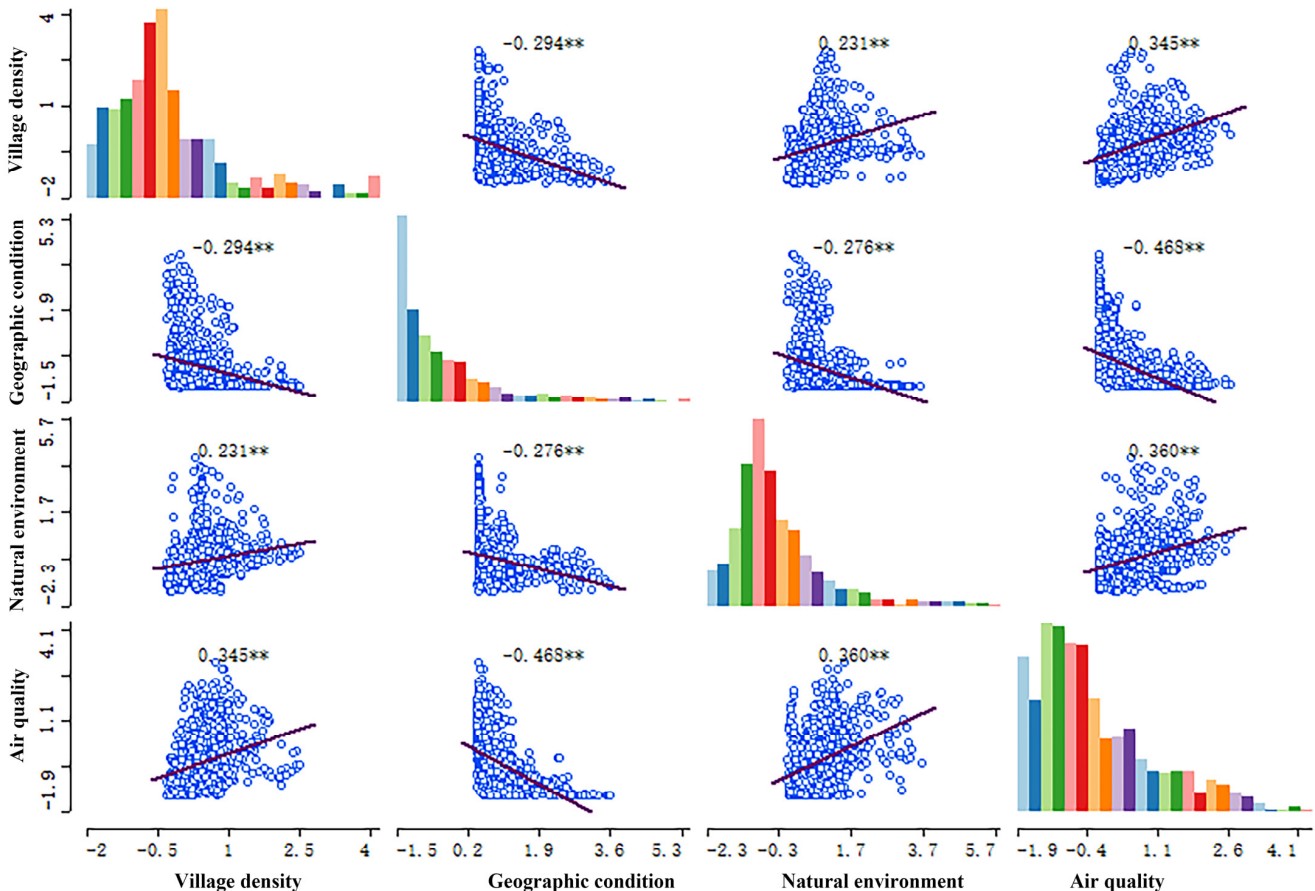

Note 1: Variables were analyzed using Pearson's correlation coefficient; Note 2: ** *p* < 0.001; Note 3: Histogram columns describe the distribution and shape of each variable.

**Figure 6.** Multivariate correlation between natural dimension and village density.

Figure 7 visually depicts the correlation between the social dimension and the spatial distribution of key villages. The investigation uncovered a relatively weaker correlation between village density and influential factors such as tourism development (0.249), economic development (0.233), economic vitality (0.196), and economic potential (0.214). In contrast, tourism potential (0.492) displayed a more prominent influence on village density compared with the other dimensions. The research outcomes suggest that tourism potential plays a pivotal role in determining the placement of villages. This observation can be attributed to the propensity of villages situated in areas characterized by significant tourism potential to attract heightened tourist activity, thus promoting economic development and growth.

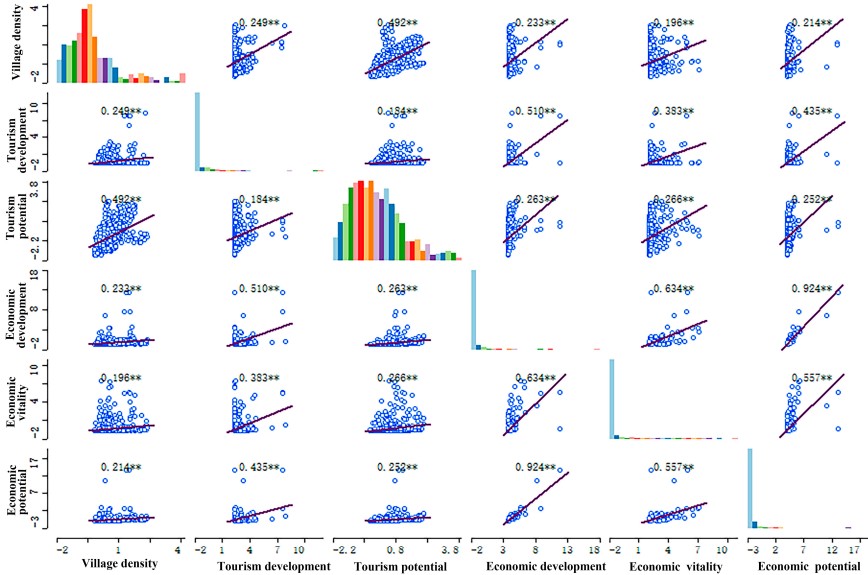

Note 1: Variables were analyzed using Pearson's correlation coefficient; Note 2: ** *p* < 0.001; Note 3: Histogram columns describe the distribution and shape of each variable.

**Figure 7.** Multivariate correlation between social dimension and village density.

Furthermore, Figure 8 presents a comprehensive overview of the correlation between the rural dimension and the spatial distribution of key villages. Within the analytical framework that examines the interrelationships among dimensions in rural areas, a relatively low correlation is observed between livelihood diversity (0.064) and village density. In contrast, factors such as rural accessibility (0.565), cultural resources (0.831), and tourism resources (0.748) exhibit a stronger influence on village density. This can be attributed to the fact that villages situated in regions with easy access, rich cultural heritage, and enticing tourism attractions are more likely to attract residents, fostering tourism growth and development.

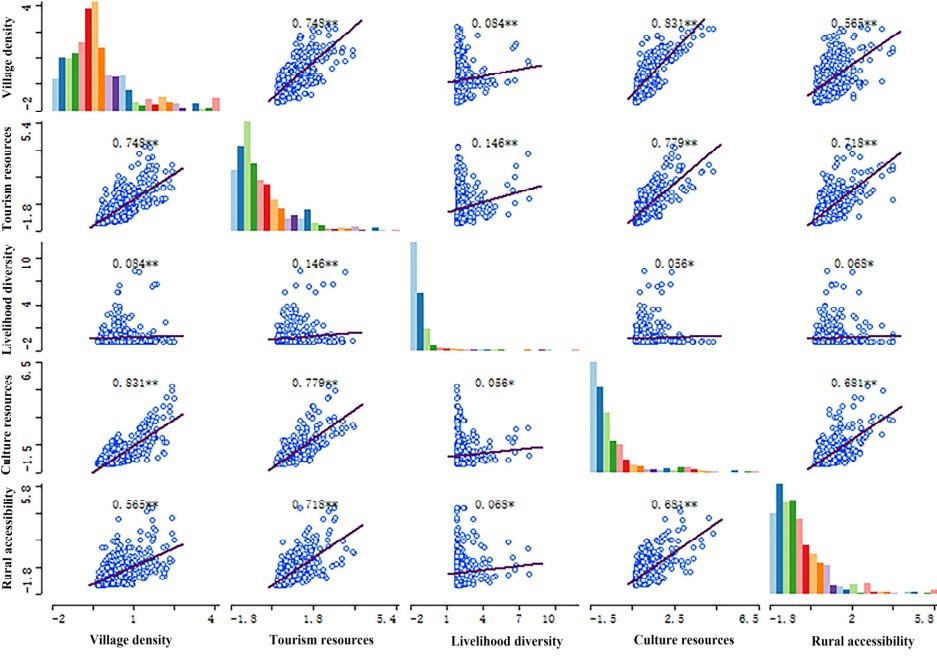

Note 1: Variables were analyzed using Pearson's correlation coefficient; Note 2: * *p* < 0.05, ** *p* < 0.001; Note 3: Histogram columns describe the distribution and shape of each variable.

**Figure 8.** Multivariate correlation between rural dimension and village density.

### 4.3. Relative Importance of Influential Factors

In this study, the random forest algorithm was employed to conduct regression analysis, aiming to assess the relative influence of various factors on village density. The outcomes of this analysis, as illustrated in Figure 9, indicate that cultural resources (1.67), tourism resources (0.9), tourism potential (0.47), and rural accessibility (0.34) emerge as the most significant determinants of village density, collectively contributing to 78% of the explanatory capacity. This finding underscores the fact that key villages are situated in areas abundant in cultural heritage, appealing tourism destinations, and convenient accessibility, as they are more likely to attract tourist, thus fostering rural tourism development. Conversely, factors such as tourism development, economic vitality, livelihood diversity, geographic conditions, economic development, natural environment, air quality, and economic potential exhibit comparatively lower levels of explanatory power.

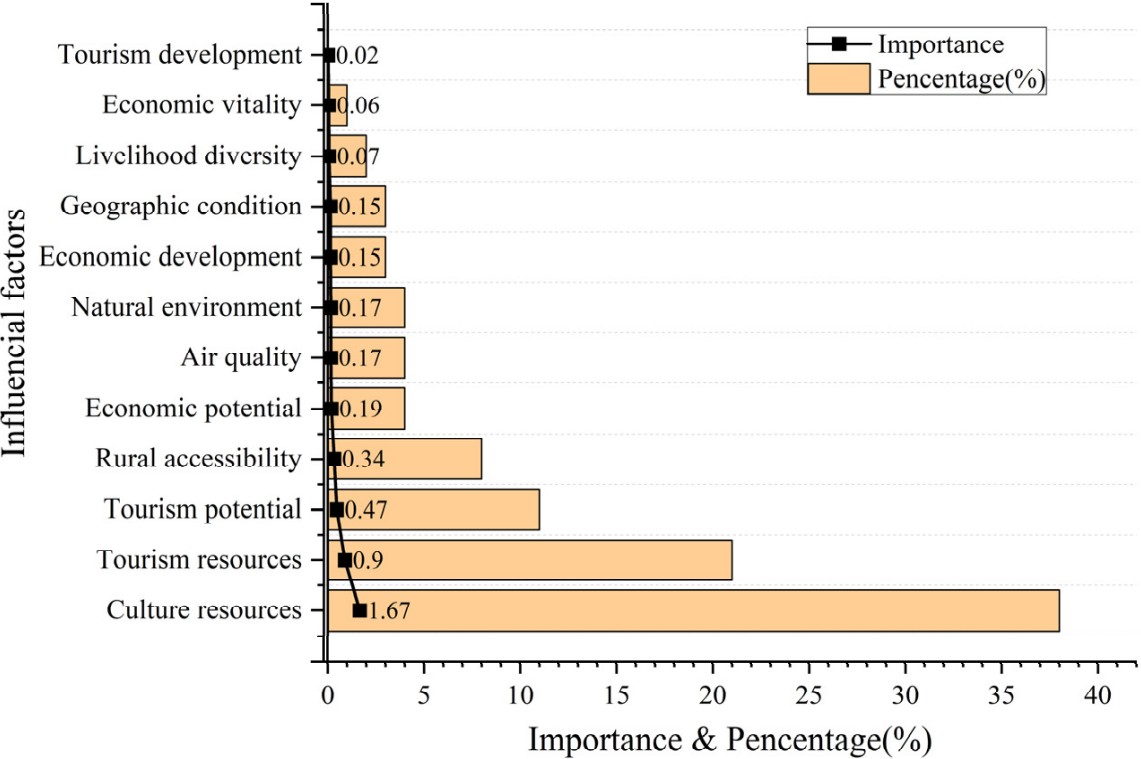

**Figure 9.** The relative importance of influential factors.

### 4.4. Potential Area of Rural Tourism Development

The primary objective of this study is to discern suitable zones for the development of rural tourism through the comprehensive analysis of pivotal factors influencing village arrangement within tourist regions. To accomplish this, a random forest algorithm was employed to establish regression models, facilitating the assessment of the suitability level for rural tourism development in alternative areas. The resultant regression outcomes pertaining to village density were classified into five distinct tiers employing the natural break method, encompassing the most suitable area, more suitable area, moderately suitable area, less suitable area, and least suitable area (Figure 10).

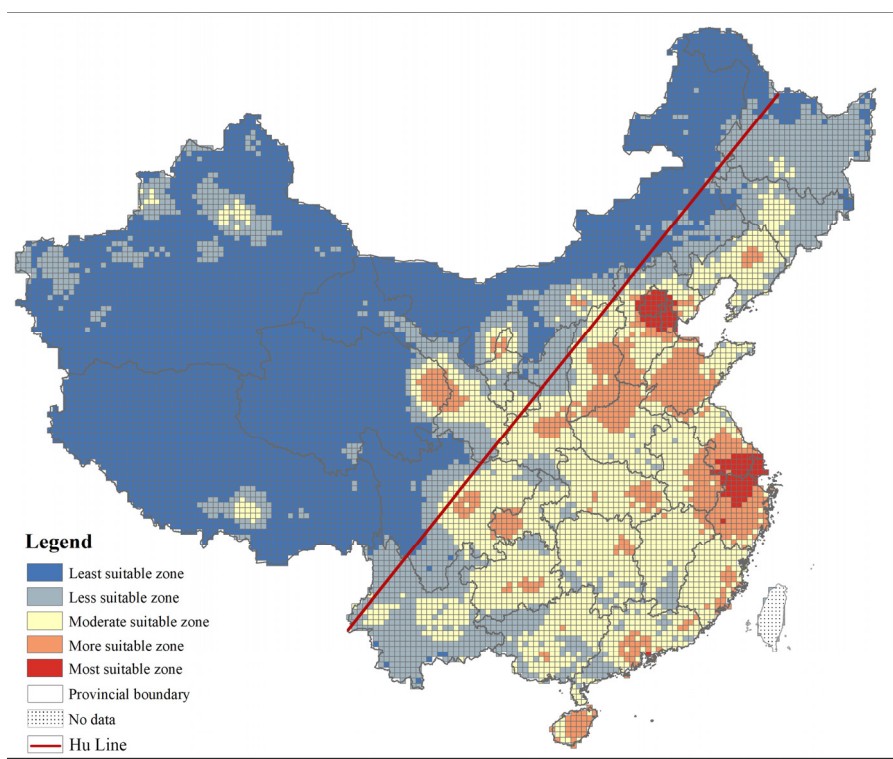

**Figure 10.** Potential area for rural tourism development in China.

In general, the areas deemed suitable for the development of rural tourism are predominantly found south of the Hu Line, where there is a high population density. The most optimal locations are situated within the urban circles of Beijing and Shanghai. Notably, Shanghai exhibits a wider range of influence, encompassing numerous comparatively suitable areas in its vicinity. Conversely, in the northern region, substantial appropriate areas are primarily distributed across Shandong, Henan, Shanxi, and Hebei provinces, owing to their dense populations and abundant cultural resources, rendering them suitable for rural tourism. Although the southern region boasts economic prosperity, this area lacks a concentrated cluster of suitable areas, resulting in their dispersal among various cities across the country. The eastern section of Hainan Island is an example of a region well suited to rural tourism. On the other hand, advancements in rural tourism are being witnessed in regions such as Sichuan, Guizhou, Hubei, Fujian, and Guangdong. Comparatively unsuitable areas tend to be dispersed along provincial boundaries and inter-provincial junctions, such as the intersection of Shaanxi, Sichuan, and Chongqing. Moreover, extensive areas in the northwest, namely Qinghai, Xinjiang and Inner Mongolia, are deemed unsuitable for rural tourism.

Interestingly, the province of Yunnan exhibits a well-established tourism industry, wherein scenic tourism predominates whereas the progress in rural tourism development remains restricted. In the western territories, the concentrated development of rural tourism is observed along the Gansu-Qinghai border, particularly in the vicinity of Xining city, as well as in the northern sector of Ningxia province. Along the Shanxi-Mongolia border, relatively suitable areas for rural tourism can be found. In Xinjiang, the relatively suitable areas are situated in the vicinity of the northern cities along the Tianshan Mountains. Overall, the border areas experience sparse population densities, thereby limiting the progress of rural tourism development.

## 5. Discussion

China attaches great importance to the development of rural tourism and considers it as a crucial means to achieve rural revitalization. The elucidation of the underlying reasons and mechanisms for the success of rural tourism in China is key to realizing its sustainable development. Although numerous scholars have analyzed the spatial patterns of rural tourism villages, beautiful rural villages, and traditional villages [16,24,25,27], providing preliminary insights into the influencing factors, there is a lack of analysis regarding the underlying mechanisms that impact rural tourism development. Therefore, this study takes key rural tourism villages as examples to uncover the spatial dynamic mechanisms driving rural tourism development, based on revealing the spatial patterns.

A comprehensive methodology is employed in this study, combining geostatistical analysis and machine learning techniques. Spatial cluster analysis, kernel density analysis, and bivariate analysis are integrated to identify clusters of key villages and examine the spatial relationships between key villages and other factors. Random forest (RF), a widely utilized machine learning method, is employed for regression analysis and prediction. It effectively assesses the relative importance of influencing factors and identifies potential areas for the development of rural tourism.

The spatial distribution of key villages in China exhibits a concentrated pattern, with a predominant presence in the eastern and central regions [16]. This distribution is notably characterized by the Hu line, which demarcates the country into two distinct economic development zones. Notably, regions such as the Yangtze River Delta, the urban circles of Beijing and Tianjin, and the middle and lower reaches of the Yellow River demonstrate a high concentration of these key villages. Rural tourism development is characterized by cluster patterns, which spread from the core regions to the periphery. In some cases, these rural tourism destinations compete with each other for tourists. Therefore, a scientific arrangement of key rural tourism villages is necessary [24].

The spatial distribution of these villages is subject to various influencing factors, encompassing natural geography, socio-economic aspects, and rural characteristics. Among these factors, the rural dimension assumes the foremost role, followed by the social dimension and natural dimension. The natural dimension pertains to the physical attributes of an area, including topography, rivers, and air quality. These factors exert an influence on rural tourism development by providing a scenic backdrop and attracting visitors. The social dimension encompasses the economic and social conditions of an area, such as population density, GDP levels, and markets. These factors impact rural tourism development by establishing a market for tourism products and services and attracting tourists. The rural dimension refers to the distinctive features of rural areas, such as cultural heritage, scenic resources, and tourism infrastructure. These factors shape rural tourism development by providing a sense of place for tourists and offering opportunities to experience a unique way of life. Villages with rich cultural heritage or unique traditions are more likely to attract tourists compared with those lacking such characteristics [25].

This study expands the existing research body by providing a comprehensive understanding of the associations between 12 influencing factors and the distribution of rural villages. Employing a comprehensive multi-factor correlation analysis, the study demonstrates significant relationships between all factors and village distribution, albeit with varying degrees of influence. Regression simulations conducted with the RF algorithm highlight cultural resources, tourism resources, tourism potential, and rural accessibility as primary determinants shaping village distribution. These findings underscore the critical role played by inherent cultural and tourism assets within rural areas, as well as their advantageous proximity to major urban markets, in facilitating rural tourism development. Additionally, the study establishes a positive correlation between enhanced rural accessibility and the advancement of rural tourism.

The assessment of rural tourism development potential emphasizes the Beijing urban circle and the Shanghai urban circle as highly favorable regions for such endeavors. Notably, the Shanghai urban circle exhibits a broader sphere of influence, extending its impact

over a wider geographic area. In the northern region, contiguous provinces with dense populations present extensive tracts suitable for rural tourism development. In contrast, the southern region features areas of moderate suitability interspersed with scattered pockets of more favorable locations. The western region, particularly the Hexi Corridor, concentrates the majority of suitable areas, whereas other regions exhibit relatively fewer opportunities. Overall, these distribution patterns of potential areas underscore the paramount importance of well-established tourism markets and abundant cultural and tourism resources in propelling rural tourism development.

## 6. Conclusions

The scientific arrangement of rural tourism is deemed as a fundamental element in achieving its sustainable development. Key villages, emblematic of China's policies aimed at revitalizing rural areas, offer insight into the spatial patterns and mechanisms that underpin the growth of rural tourism in the country. This understanding, in turn, can inform decision-making processes aimed at fostering the long-term sustainability of rural tourism.

Through the application of diverse research methodologies, this study provides valuable and illuminating findings. By pinpointing suitable areas and elucidating the determinants that shape the spatial distribution of rural tourism villages, this research offers crucial insights for policymakers and tourism developers, facilitating the promotion and advancement of rural tourism in China. In comparison to previous studies conducted at the provincial level, this research extends the investigation to a larger scale, yielding more refined outcomes. These research findings have substantial implications for regional planners and policymakers, providing valuable insights into the primary determinants influencing rural tourism development around the world. Nonetheless, this study is subject to several limitations. It is crucial to acknowledge that the presence of missing data in certain regions may impact the research outcomes when addressing data gaps. In addition, the selection of influencing factors for rural tourism is biased towards supply-side factors, with limited consideration given to demand-side factors. Furthermore, future research should consider utilizing other machine learning algorithms in addition to RF algorithms.

**Author Contributions:** Conceptualization, Xishihui Du and Zhaoguo Wang; methodology, Xishihui Du and Yan Wang; writing—original draft preparation and review, Xishihui Du, Yan Wang and Zhaoguo Wang. All authors have read and agreed to the published version of the manuscript.

**Funding:** This work was supported by the Foundation of the Educational Department of Liaoning Province (Grant No. lnjc202015 and LJKMR20221067) and National Natural Science Foundation of China (Grant No. 42101294).

**Data Availability Statement:** All data that support the findings of this study are available from the corresponding author upon reasonable request.

**Acknowledgments:** The authors thank the anonymous reviewers for their insightful comments and helpful suggestions that helped improve the quality of our manuscript.

**Conflicts of Interest:** The authors declare no conflict of interest.

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
