# Peer review of "The Spatial Mechanism and Predication of Rural Tourism Development in China: A Random Forest Regression Analysis"

_ijgi, doi:10.3390/ijgi12080321_

Round 1

Reviewer 1 Report

- Figure 1 and table 1: the data collection and variables should be better aligned with the factors reported in the literature (73-74): socio-economic, transit accessibility, tourist demand, policy orientation, service quality.

- Is policy orientation not addressed because it is part of the selection of key rural tourism villages? If so, that is acceptable justification for not further analyzing it, yet it should be mentioned.

- 138: explain how the selection of key villages was conducted and identify the consequences on the key factors represented by the data.

- table 1 and 178-179. Cultural ecosystem services include benefits from natural elements. Why take them as a separate factor? It looks like the RF method is an alternative for calculating cultural ecosystem services. This is very interesting, because there is a need for methods to quantify these ecosystem services.

- Title 2.2.2 and caption Figure 3. : these are the natural, societal and rural characteristics of the key rural tourism villages. It is confusing to use the term 'influential factors'

- 3.1 and 3.2 are OK, yet very classic analyses. Maybe this could be shortened?

- The RF algorithm is an interesting way of testing the potential for tourism development. Yet, it looks like a complex calculation to come up with straightforward conclusions. The article would benefit from a critical assessment of the advantages and disadvantages of the method (e.g. potential overfitting, low prediction accuracy compared to other machine learning algorithms, ...), thus bringing more nuance in the conclusions

Reviewer 2 Report

 Review of article: The spatial mechanism and predication of rural tourism devel-2 opment in China: A random forest regression analysis

 The article aim and main findings fits to the content of “International Journal of Geo-Information” journal.

 The main aim of research and main findings are presented in the Abstract. Its good that main method of research - random forest analysis was shortly mentioned.

However, I miss shortly mentioned problem – why such research was done. Now all abstract mainly describes findings and conclusions.

 Introduction is wide enough describing the problems and importance of rural tourism in China. Wide description of rural tourism types were overviewed and presented.

The methods of research are presented and explained very clearly. The process of data gathering and analysis is shown very clearly. Influential factors were described and explained.

However, maybe there are more factors which influence rural tourism but they were not considered – why?

Results section.

Spatial distribution characteristics such as spatial clusters of key rural tourism villages and spatial distribution density of key rural tourism villages were determined. Correlation between natural dimension, social dimension and village density were calculated and shown. The relative importance of influential factors were discussed.

Further potential areas for rural tourism development were identified.

 Its a very nice and interesting article, I am grateful for authors.

Reviewer 3 Report

This study focuses on key tourism villages and employs geostatistical analysis and the random forest methodology to elucidate the spatial mechanisms underlying rural tourism and identify potential areas for its development in China. The study seems interesting and meaningful. However, several aspects need more details and I would like to share with my specific comments and suggestions below:

1. A deep literature review should be given. RF is a popular method used in many field, relevant literature should be mentions. E.g. “10.1007/978-3-319-06483-3_9; 10.1016/j.jag.2021.102475; 10.1109/TGRS.2021.3080083.

2. Line 150 to 152 in section 2.2.2, it is not clear why the factors listed in Table 1 are the most significance and relevance features for this study.

3. In section 2.3.1, what is the purpose and object of using HDBSCAN clustering?

4. In section 3.1.1, 26 distinct clusters were generated by HDBSCAN algorithm, but how to ensure the rationality of the clustering results.

5. Line 354 to 355 in section 3.5, what specific method is used to calculate the importance should be given a more detailed explanation.

Specific details:

1. The seventh figure in Figure 3 shows that the color of the map boundary line is inconsistent with other maps.

2. The legend of the red line is missing in Figure 5 and 10.

3. The legend of the histograms is missing in Figure 6 and 7.

There are some typos and grammatical mistakes in the paper. Please check it carefully.

Reviewer 4 Report

Thank you for the opportunity of reading and reviewing your manuscript.
The paper is interesting, it deals with the important topic of the
 spatial mechanism and predication (prediction?) of rural tourism development in China and it is my pleasure to review it.

The paper is very detailed and uses a solid scientific and logical tool. Methodology and approaches are interesting, systematic and comprehensive.

However, I would have some considerations and suggestions for improving the quality of the article.

At Row 46 the statement "China is predominantly an agricultural country" is confusing, as the latest statistical data indicate the agriculture (including forestry, fishing) share in China’s GDP is approx. 8% of GDP. The Introduction Chapter is quite long, with a lot ofvarious information and contextual approaches. It is recommendable to insert a literature review section/chapter - this information already exists in Introduction chapter, but it is scattered and poorly systematized. Thenit follows a description of the actual situation, preparing the transition to the Methodology chapter; Similarly, the Chapter 4.4. should be divided into 2 chapters: Discussion, included or not in Chapter 4, and, respectively, Conclusions, - a new chapter that systematizes and summarizes the main research results, policy implications, future research directions (as an invitation to an academic and professional debate on the future research topics); What are the working hypotheses or research questions? Stating them would help a lot in structuring and understanding the purpose of the research and guiding the interested readers; The paper focuses, as expected, on the elements of the tourist offer (implicitly the natural background) and finally makes recommendations, highlighting the advantages, potential and opportunities offered by the different regions/areas, etc. However, there are few analyses coming from the demand side, consumer preferences and expectations, consumption trends, social choice mechanisms, opportunity costs, etc. Of course, the collection and processing of this information would go far beyond the scope of the paper, but mentioning them would emphasize the openness of the authors to enrich present and further researches on this topic, encompassing various perspectives (economic, social, psychological). Even if the main results are circumscribed to a contextual framework, remarkable for its diversity and extent (China), it would be recommendable for the authors to suggest the (possible) relevance of the present research for other contexts, for various developing countries which are endowed with remarkable natural and rural resources that can be exploited in domestic/international tourism, contributing to the economic growth of the respective countries/regions/local communities. Also, we recommend inserting a paragraph stating the main limitations of the paper.

Thank you for the opportunity to review this article and good luck!

Round 2

Reviewer 3 Report

I have no more problems.

Reviewer 4 Report

Thank you for providing the revised version of your paper. My main comments were addressed.